# Characterization of the Soil Bacterial Community from Selected Boxwood Gardens across the United States

**DOI:** 10.3390/microorganisms10081514

**Published:** 2022-07-26

**Authors:** Xiaoping Li, Ping Kong, Margery Daughtrey, Kathleen Kosta, Scott Schirmer, Matthew Howle, Michael Likins, Chuanxue Hong

**Affiliations:** 1Hampton Roads Agricultural Research and Extension Center, Virginia Tech, Virginia Beach, VA 23455, USA; pkong@vt.edu (P.K.); chhong2@vt.edu (C.H.); 2Long Island Horticultural Research and Extension Center, Cornell University, Riverhead, NY 11901, USA; mld9@cornell.edu; 3California Department of Food and Agriculture, Sacramento, CA 95814, USA; katfish@frontiernet.net; 4Bureau of Environmental Programs, Illinois Department of Agriculture, Dekalb, IL 60115, USA; scott.schirmer@illinois.gov; 5Department of Plant Industry, Clemson University, Florence, SC 29506, USA; mhowle@clemson.edu; 6Chesterfield Cooperative Extension, Chesterfield County, VA 23832, USA; likinsm@chesterfield.gov

**Keywords:** disease suppressive soil, soil bacterial community, urban garden, boxwood, biological control agents, Nanopore MinION sequencing

## Abstract

In a recent study, we observed a rapid decline of the boxwood blight pathogen *Calonectria pseudonaviculata* (*Cps*) soil population in all surveyed gardens across the United States, and we speculated that these garden soils might be suppressive to *Cps*. This study aimed to characterize the soil bacterial community in these boxwood gardens. Soil samples were taken from one garden in California, Illinois, South Carolina, and Virginia and two in New York in early summer and late fall of 2017 and 2018. Soil DNA was extracted and its 16S rRNA amplicons were sequenced using the Nanopore MinION^®^ platform. These garden soils were consistently dominated by *Rhizobiales* and *Burkholderiales*, regardless of garden location and sampling time. These two orders contain many species or strains capable of pathogen suppression and plant fitness improvement. Overall, 66 bacterial taxa were identified in this study that are known to have strains with biological control activity (BCA) against plant pathogens. Among the most abundant were *Pseudomonas* spp. and *Bacillus* spp., which may have contributed to the *Cps* decline in these garden soils. This study highlights the importance of soil microorganisms in plant health and provides a new perspective on garden disease management using the soil microbiome.

## 1. Introduction

Bacterial communities are important components of soil health in general and soil suppressiveness in particular. Soil disease suppression refers to the capacity of a soil that maintains low disease severity or incidence despite the presence of pathogen inoculum and conducive conditions [1,2,3]. There is an increasing body of evidence recognizing the importance of microorganisms in soil disease suppression [4,5,6,7,8,9]. For example, fluorescent *Pseudomonas* spp. are key players in the decline of take-all disease by producing the antibiotic 2,4-diacetylphloroglucinol [10]. The populations of *Burkholderia* and *Streptomyces* were enriched in healthy banana rhizosphere soils compared with those infested with *Fusarium* species [11].

Soil suppression can be either general or specific in terms of soil microorganism interaction with target pathogens [10]. General suppression refers to soil microorganisms as a whole competing with pathogens for nutrients and niche habitats [10,12]. It implies the collective and non-discriminating activities of soil microbial communities against pathogens. The microbial composition of such suppressive soils is distinct from that of disease conducive soils [8,11] or the microbial diversity is positively correlated with disease suppression [13]. Specific suppression refers to certain soil microorganisms being antagonistic against pathogens [10,14]. These antagonistic microbes or biocontrol agents (BCAs) protect plants through various mechanisms, including antimicrobial compounds, competition, hyper-parasitism, and/or induced plant resistance [15]. Hence, soil microbial community features and certain antagonistic microbial populations could be indicative of soil suppressiveness [16,17].

Among the most studied soil microbial taxa are bacteria in agricultural crop systems [18,19,20,21]. Many bacterial species and strains from the genera *Bacillus*, *Pseudomonas*, and *Streptomyces* have been shown to have biological control activities against a wide range of fungal plant pathogens and have been subsequently developed into biofungicides for disease management [22]. There have also been studies examining how farming practices, such as soil organic amendment and crop rotation, may change soil bacterial communities and consequently improve soil suppressiveness and plant productivity [23,24,25,26]. Soil suppressiveness is also transferable, alleviating disease pressure in treated soils, although it depends on many biotic and abiotic factors in the soil environment [9,27,28,29].

Comparatively, the garden soil microbiome has been largely neglected. One of the most comprehensive studies was recently done by LeBlanc and Crouch, sampling soil of 82 individual curated boxwood accessions at the U.S. National Arboretum, demonstrating that bacterial diversity was significantly different in soil from distinct types of boxwood [30]. Although this work was done in the absence of boxwood disease, it demonstrated the potential for improving boxwood health by planting different species or cultivars in the landscape to manipulate the soil microbiome. In a more recent study, we demonstrated that the soil population of *Calonectria pseudonaviculata* (*Cps*), a destructive fungal pathogen of boxwood crops and plantings [31,32], declined sharply within the first year of blighted boxwood removal and fell to an almost undetectable level at the end of year 2 or 3 in all selected gardens across the United States [33]. We speculated that soil bacteria in these gardens may have contributed to the *Cps* decline, as has been shown for other plant pathogens [34].

The primary objective of this study was to characterize the bacterial communities in these garden soils using the MinION^®^ platform (Oxford Nanopore Technologies, Oxford, UK). Specifically, we profiled the bacterial communities, identified beneficial members that included potential pathogen antagonists, and evaluated their variation across gardens and sampling times.

## 2. Materials and Methods

### 2.1. Boxwood Gardens

As described previously [33], private gardens included in this study represented five geographic regions of the United States: California (San Mateo County) for Pacific West, Illinois (Cook County) for North Central, New York (Long Island) for Northeast, South Carolina (City of Florence) for Southeast, and Virginia (Powhatan County) for the Mid-Atlantic region. Five gardens, one in each state, were sampled four times, twice per year in 2017 and 2018. A second New York garden also on Long Island was sampled twice, once per year in 2017 and 2018, and included in the analysis for New York samples.

### 2.2. Soil Sampling

Soil samples were collected in three replicates from individual gardens in early summer (ES) and late fall (LF) of 2017 and 2018 using the same protocol in a coordinated fashion [33]. Briefly, the top 6 cm of soil including limited leaf debris was taken using a soil sampler and placed in a new Ziploc^®^ bag (Bay City, MI, USA). Virginia samples were transported in a cooler to the lab, while samples of non-Virginia origin were sent overnight via a commercial carrier to Virginia Tech for processing and analyses. The soil samples were placed at 4 °C for short-term storage or −80 °C for long-term storage.

### 2.3. DNA Extraction and PCR Amplification

The soil samples were first equilibrated to room temperature before DNA extraction. Because moisture levels were different among samples, volume was used instead of weight to facilitate comparison across locations. For each replicate sample, 0.4 cc of soil was used for DNA extraction. Soil DNA extraction was carried out using Qiagen PowerLyer PowerSoil kits (Qiagen, Germantown, MD, USA) according to the manufacturer’s protocol with a few modifications. Specifically, soil was first added to a PowerBead Pro tube filled with 750 µL of PowerBead solution and 60 µL of Solution C1. Second, DNA was cleaned using the Maxwell RSC cartridge (Promega Mad-74, Madison, WI, USA) for automation and eluted with 60 µL of the elution buffer. DNA concentration was determined using the QuantiFluor ONE system (Promega, Madison, WI, USA).

The 16S rRNA gene was used as the DNA marker to identify bacterial members in each sample and was amplified using primer pair 27F (5′-AGAGTTTGATCCTGGCTCAG-3′) and 1492R (5′-GGTTACCTTGTTACGACTT-3′) [35]. The primer pair was attached to the tail of the ONT overhand sequences (5′-TTTCTGTTGGTGCTGATATTGC-project specific forward primer sequence-3′, 5′-ACTTGCCTGTCGCTCTATCTTC-project specific reverse primer sequence-3′). The PCR conditions were 94 °C for 2 min, followed by 30 cycles of 94 °C for 30 s, 65 °C for 45 s, and 72 °C for 1 min, and then 72 °C for 10 min. PCR products were cleaned using magnetic beads from the MagBio HighPrep^TM^ PCR protocol (MagBio Genomics Inc., Gaithersburg, MD, USA).

### 2.4. Nanopore Library Preparation and Sequencing

Multiplex Nanopore libraries were prepared using an SQK-LSK 109 ligation kit (Oxford Nanopore Technologies, Oxford, UK). Amplicons of the DNA fragments for bacteria were pooled in 0.5 µg. Fifty fmol samples were used and the volume was adjusted to 48 µL with nuclease-free water. A PBC001 barcode kit (Oxford Nanopore Technologies, Oxford, UK) and the LongAmp^®^ Tag 2x master mix (New England Biolabs, Ipswich, MA, USA) were used to barcode each sample. Thermal conditions for barcoding followed the ONT protocol. Twelve barcoded samples were then subjected to cleanup with MagBio HighPrep^TM^ beads and pooled in equal molar concentration of 100 fmol per sample to a final of 1 µg in 47 µL nuclease-free water. The barcoded DNA library was prepared for ligation using the NEBNext^®^ FFPE DNA Repair module and NEBNext^®^ Ultra II End Repair module (New England Biolabs, Ipswich, MA, USA). Following cleanup with AMPure XP beads (Beckman Coulter Life Sciences, Indianapolis, IN, USA), the library was ligated to the ONT adapters with NEBNext Quick T4 ligase (New England Biolabs, Ipswich, MA, USA). Short fragment buffer (SFB) (Oxford Nanopore Technologies, Oxford, UK) was selected in the following cleanup to retain all fragments. The library was then quantified and approximately 50 µg was loaded into a Nanopore MinION R9.4 flow cell following the ONT priming and loading protocol. Fast basecalling was selected in the MinKNOW software (GUI version 3.4.5, Oxford Nanopore Technologies, Oxford, UK), and the Nanopore proprietary software Guppy (CPU version 3.0.4) was also installed and coupled with MinKNOW to facilitate basecalling on a Windows 10 system.

### 2.5. Database Construction

The SILVA database and the complete bacterial genome database from the National Center for Biotechnology Information (NCBI) were used separately for chimera removal and taxonomy assignment. The SILVA 138.1 SSU NR99 [36,37] was downloaded from the official repository. The raw reference sequences and taxonomy information were processed in the quantitative insights into microbial ecology (QIIME2) [38] environment with the REference Sequence annotation and CuRaIon Pipeline (RESCRIPt) [39] for curation. Briefly, the low-quality reference sequences, i.e., those with ambiguous bases (default minimal 5) and homopolymers (default minimal 8), were first removed. The reference sequences were then filtered by length to retain a minimum length of 1200 bp for the SILVA database. Dereplication followed to keep the unique sequences with different taxonomies (set p-mode “uniq”). After curation, the SILVA database contained 438,119 sequences, representing 90 bacterial phyla, 245 bacterial classes, 657 bacterial orders, 1203 bacterial families, and 4333 bacterial genera, or 23,511 bacterial species. The software centrifuge was used to compile the reference database for taxonomy assignment, retrieved automatically from the NCBI complete genome database [40].

### 2.6. Bioinformatics

An in-house Python package (Nanoprep, https://github.com/xpli2020/NanoPrep, accessed on 20 October 2021) was developed to facilitate the implementation of various bioinformatics tools and file navigations. All quality fastq reads (≥Q 10) were first demultiplexed by the attached barcodes using qcat (Oxford Nanopore Technologies, Oxford, UK). The NanoPlot [41] program was then used to visualize the quality of all reads. The NanoFilt function [41] filtered the reads with the minimum quality set at 10 and the length between 1000 bp and 2000 bp (-l 1000, -maxlength 2000, -q 10, -headcrop 50, -tailcrop 50). The fastq reads were converted to fasta using seqtk (https://github.com/lh3/seqtk, accessed on 20 October 2021). Minimap2 [42] and Yacrd [43] were used to remove chimeras. The “clean” reads were then re-aligned to the reference database using Minimap2. The reads with the best alignment score were retained. An operational taxonomic unit (OTU) table was constructed based on the alignment output of Minimap2 using custom R scripts. The OTU table, the taxonomy table, and the metadata were all imported into an R environment [44] for downstream statistical analyses using the phyloseq package [45].

### 2.7. Data Analyses and Statistics

Kingdoms and phyla that contained “NA” values and non-bacteria were removed. Taxa that contained more than 10 reads and samples that contained more than 20,000 reads were retained to reduce bias from the singletons or error reads. Sample coverage was calculated using the metagMisc package [46] with singleton correction [47]. The significance level for all statistical tests was set at 5%.

#### 2.7.1. Community Diversity and Structure

Observed OTU and Shannon’s index were used to estimate the alpha diversity with the estimate_richness function from the phyloseq package [45]. Samples were rarefied at 24,400 reads. Two-way analysis of variance (ANOVA) was used to assess season and location effects on the alpha diversity indices, followed by mean separation according to Tukey honest significant difference (HSD) at *p* = 0.05. Effect size was evaluated using the anova_stats function from the sjstats package [48]. Small (η^2^ = 0.01), medium (η^2^ = 0.06), and large (η^2^ = 0.14) effect sizes were defined by Cohen [49].

Bray–Curtis dissimilarity [50] was used to analyze the community structure, and the visualization of the matrix was based on the principal coordinates analysis (PCoA) ordination. To determine which factor drove the community structure in 2017 and 2018, the Bray–Curtis dissimilarity was analyzed using the adonis function of the vegan package [51] with 10,000 permutations: Bray–Curtis dissimilarity ~ State + Season + State × Season. Dispersion was evaluated using the Vegan betadisper function and the statistics were obtained from an ANOVA test.

#### 2.7.2. Identification of Major Bacterial Orders

The OTU table was first agglomerated to the order level. Relative abundance was calculated, and the bar plot was produced using ggplot2 [52]. All bacterial orders were further analyzed individually using the Kruskal–Wallis test [53] to assess the level of variation in their abundance among five states and between two seasons. *p*-values were adjusted using the false discovery rate (FDR) procedure.

#### 2.7.3. Identification of Bacterial Species with Biological Control Potential

This began with compiling a list of bacterial species known to have strains with antagonistic activities against plant pathogens based on literature reviews [54,55,56,57,58] and the SMARTBIOCONTROL database (http://www.smartbiocontrol.eu/en/database-effects-of-biocontrol-agents/, accessed on 10 May 2022). The OTU table was first agglomerated to the species level and then evaluated against the list to identify bacterial species known to have biological control potential in sampled garden soils. Subsequently, the most abundant species were analyzed individually using the Kruskal–Wallis test to assess the level of variation in their abundance among soil samples from five states and between two seasons. *p*-values were adjusted using the FDR procedure.

## 3. Results

### 3.1. Sequencing Summary

A total of 32,129,795 raw reads were generated in eight runs using five nanopore MinION R9.4 flow cells. After demultiplexing using qcat, approximately 30 million quality reads were obtained, averaging 4.3 million per run. With further filtering using Nanofilt, eight million reads were retained, averaging about 1.2 million reads per run (Appendix A). The quality of the sequencing reads was at least 10, suggesting the accuracy was over 90%. After further filtering (the minimum number of reads for each sample was at least 20,000 and minimum number of reads for each taxon was at least 10) in an R environment, about 7.5 million reads were retained (Table 1). Most of the samples reached a reasonable sequencing depth, although not all were saturated (Appendix A).

### 3.2. Garden Soil Bacterial Community Diversity

A total of 1,788,962 reads were identified for samples collected in early summer with sample coverage at 0.991, while 2,127,058 reads were identified for late fall in 2017 with sample coverage at 0.991 (Table 1). Likewise, 1,871,473 reads were identified for the samples collected in early summer with sample coverage at 0.995, while 1,732,272 were identified for late fall in 2018 with sample coverage at 0.990 (Table 1). Overall, the sample average was 0.992. The observed richness and Shannon index in the bacterial community varied significantly among the five states (*p* < 0.0001) and between the two seasons (Table 2 and Figure 1). The only exception was that the Shannon index did not differ between early summer and late fall samples of 2018 (Table 2). The lowest OTU number was observed in Virginia samples across all sampling times from early summer of 2017 to late fall of 2018 (Figure 1). The greatest variation in OTU number was observed in the New York samples with the highest in late fall of 2018 (*p* = 0.0014) and the lowest in early summer of 2017 (Appendix A). The second greatest variation was seen in the California samples, followed by those collected from South Carolina. The most consistent OTU richness across all sampling times was observed in the Illinois samples. Similarly, the most variation in the Shannon index was observed in the New York samples in 2017 (*p* = 0.0247) and 2018 (*p* = 0.0279) (Appendix A).

### 3.3. Garden Soil Bacterial Community Structure

Soil microbial community structure was measured using the Bray–Curtis dissimilarity index and ordinated for visualization using principal coordinates analysis (PCoA). Overall, the first axis explained 26.1% and 25.5% of the variation in 2017 and 2018, respectively (Figure 2). VA samples appeared separated from those collected from other states (Figure 2 and Appendix A). The centroids of IL, CA, and NY were closely gathered and samples from those states overlapped in 2017 (Appendix A), while the centroids spread out in 2018, with CA samples separate from the rest (Appendix A). In contrast, samples annotated by season were overlapped in both 2017 and 2018 (Figure 2). The PERMANOVA test on the Bray–Curtis dissimilarity indicated a strong state effect on the dissimilarity variances in both 2017 (*p* < 0.0001) and 2018 (*p* < 0.0001) (Table 3). A significant season effect also was observed in the bacterial community structure, accounting for 6% and 5% of the variance in 2017 and 2018, respectively (Table 3). Additionally, strong interactions between state and season were observed in the community structure, accounting for 22% and 13% of the total variation in 2017 and 2018, respectively (Table 2).

### 3.4. Relative Abundance of Bacterial Orders in Sampled Garden Soil

The ten most abundant bacterial orders in these garden soils were *Rhizobiales*, accounting for 16.8% of the relative abundance, followed by *Burkholderiales* at 16.0%, *Vicinamibacterales* at 6.9%, *Chitinophagales* at 3.8%, *Xanthomonadales* at 2.8%, *Gaiellales* at 1.9%, *Sphingomonadales* at 1.8%, *Bacillales* at 1.8%, *Acidobacteriales* at 1.7%, and *Gemmatimonadales* at 1.6% (Table 4).

Similar bacterial dominance was observed in both sampling seasons. In the early summer of 2017, the most abundant order was *Rhizobiales* at 16.0%, followed by *Burkholderiales* at 14.8%, *Vicinamibacterales* at 5.5%, *Chitinophagales* at 5.1%, *Xanthomonadales* at 4.4%, *Acidobacteriales* at 3.8%, and *Bacillales* at 3.0% (Figure 3a). In the late fall of 2017, the most abundant order was *Burkholderiales* at 17.0%, followed by *Rhizobiales* at 16.7%, *Vicinamibacterales* at 8.2%, and *Chitinophagales* at 3.2% (Figure 3b). In the early summer of 2018, among the most abundant orders were *Burkholderiales* at 17.3%, *Rhizobiales* at 15.5%, *Vicinamibacterales* 6.3%, *Chitinophagales* 4.0%, and *Xanthomonadales* at 3.5% (Figure 3c). In the late fall of 2018, among the most abundant orders were *Rhizobiales* at 19.1%, *Burkholderiales* at 14.6%, and *Vicinamibacterales* at 7.5% (Figure 3d). Notably, *Rhizobiales* and *Burkholderiales* were the two orders with over 10% relative abundance across all years, seasons, and states (Appendix A). The relative abundance of the other eight orders varied with state and season from 12.9% to 1.9% in 2017, and from 11.4% to 2.0% in 2018 (Appendix A).

The relative abundance of identified bacterial orders varied with state, but not with season (Table 4 and Figure 3). Overall, the relative abundances of 226 and 209 bacterial orders (total = 397) were significantly affected by state in both 2017 and 2018 (Table 4 and Figure 3).

### 3.5. Bacterial Species with Biological Control Potential

A total of 66 bacterial species known to have strains with biological control potential (BCA candidates) were identified from these garden soils, and the diversity of these BCA candidates differed among the states, but not between two seasons within each year and state. Specifically, the richness and the Shannon’s index were different among the states in 2017 (*P*_Richness_ = 0.0033, *P*_Shannon_ = 0.0396) and 2018 (*P*_Richness_ = 0.0004, *P*_Shannon_ < 0.0001). The interaction between state and season was also significant for the richness and the Shannon’s index in 2018 (*P*_Richness_ = 0.0439, *P*_Shannon_ = 0.0002). Comparatively, the location effect (η^2^) was large for the richness (η^2^_2017_ = 0.367, η^2^_2018_ = 0.418) and Shannon’s index (η^2^_2017_ = 0.245, η^2^_2018_ = 0.502), while the interaction effect was medium to large: 0.097 and 0.163 for the richness and 0.127 and 0.266 for the Shannon’s index in 2017 and 2018, respectively.

Together, the 66 BCA candidates accounted for 1.4% of the total bacterial species (*n* = 4636) identified in this study. The most abundant species were *Pseudomonas* sp. at 0.32%, *Bacillus* sp. at 0.31%, *Rhizobium* sp. at 0.21%, *Lysobacter* sp. at 0.16%, *Paenibacillus* sp. at 0.15%, and *Arthrobacter* sp. at 0.10% (Figure 4).

The BCA candidates identified belonged to 12 orders (Appendix A) and 21 genera (Appendix A). At the order level, *Pseudomonadales* accounted for 24.8% (*n* = 2400), followed by *Bacillales* at 19.4% (*n* = 1875), *Enterobacterales* at 10.9% (*n* = 1050), *Rhizobiales* at 10.1% (*n* = 975), *Xanthomonadales* at 7.8% (*n* = 750), *Streptomycetales* at 7% (*n* = 675), *Paenibacillales* at 6.2% (*n* = 600), and *Burkholderiales* at 4.7% (*n* = 450) (Appendix A). At the genus level, *Pseudomonas* was at 36.4% (*n* = 24), followed by *Streptomyces* at 12.1% (*n* = 8), *Bacillus* at 9.1% (*n* = 6), *Pantoea* at 6.1% (*n* = 4), *Paenibacillus* at 4.5% (*n* = 3), and *Burkholderia* at 3% (*n* = 2) (Appendix A).

In general, the 66 BCA candidate species were distributed evenly among the states (Table 5). The average relative abundances were 2.03% and 1.09% in ES and LF seasons of 2017, respectively, and 2.14% and 1.87% for the same sampling times of 2018. Their relative abundances were generally consistent among the five states and between the two seasons in 2017 with a few exceptions. Seasonal differences were observed in *Bacillus* sp. (*p.*adj = 0.0001), *Rhizobium* sp. (*p.*adj = 0.0361) (Figure 4a), and *Pantoea agglomerans* (*p.*adj = 0.0361, data not shown). Location differences were seen in *Bacillus* sp. (*p.*adj = 0.0001) (Figure 4a) and *Lysobacter enzymogenes* (*p.*adj = 0.0412, data not shown). Variations were observed in the relative abundance of 26 BCA candidates among the states/gardens in 2018, including the most abundant *Pseudomonas* sp. (*p.*adj = 0.0020), *Bacillus* sp. (*p.*adj < 0.0001), *Rhizobium* sp. (*p.*adj = 0.0066), and *Arthrobacter* sp. (*p.*adj = 0.0001) (Figure 4b). No seasonal variation was observed in 2018.

## 4. Discussion

This study characterized the bacterial communities in urban garden soils where the boxwood blight pathogen *Calonectria pseudonaviculata* (*Cps*) population declined rapidly [33]. We reported two important discoveries. First, *Rhizobiales* and *Burkholderiales* were the dominant bacterial orders across all five states/gardens and the two sampling times in both 2017 and 2018. Second, 66 bacterial species known to have strains with antagonistic activities against plant pathogens (BCA candidates) were also identified. The bacterial orders and BCA candidates observed resemble those in the suppressive soils from other environments [21,59,60,61,62]. They are important components of a healthy soil microbiome [3,4,63,64], and likely have contributed to the decline of the *Cps* population in these garden soils [33]. These results highlight the importance of microbiome components in garden soil health and provide a new perspective from which to undertake ornamental plant disease management in gardens and other public spaces in the future.

There are several lines of evidence supporting that the most abundant bacterial orders and BCA candidate species may have contributed to the decline of the boxwood blight pathogen population in all of the gardens surveyed [33]. First, *Burkholderiales*, one of the most dominant orders identified in this study, is known to have species and strains with biological control and other beneficial activities. Specifically, *Burkholderia* strain SSG isolated from boxwood has been demonstrated to be highly effective as a biocontrol agent against boxwood blight [65] and other diseases caused by a variety of pathogens including bacteria, oomycetes, other fungi, and one virus [66], as well as acting as a biofertilizer and plant defense inducer [65,67]. These broad-spectrum biological activities are supported by the many antibiosis genes and clusters on the *Burkholderia* sp. SSG genome [68]. The identification of another predominant order, *Rhizobiales*, a key taxon in the boxwood soil microbial network [30], suggests that this mostly root-associated bacterial order may be specifically recruited by boxwood to carry out multiple functions, such as nitrogen-fixing and root growth promotion [69,70]. Third, other BCA candidates including some well-known genera are also dominant. For instance, *Pseudomonas* spp., *Bacillus* spp., *Rhizobium* spp., *Lysobacter* spp., *Paenibacillus* spp., and *Arthrobacter* spp. have all been reported extensively for their capacity to control plant pathogens [71,72,73,74,75,76,77,78,79]. Specifically, *Pseudomonas*, *Bacillus*, *Paenibacillus*, and *Streptomyces* species are well known to inhibit the formation and germination of microsclerotia by *Verticillium* spp. or increase their mortality [80,81,82,83], by producing volatile organic compounds [81] or chitin lysis enzymes [62,79]. Some of the same genera identified in this study likely have contributed to the decline of *Cps*, which also produces microsclerotia for survival in soils [84]. Fourth, all of the BCA candidates found in this study were rather abundant in these garden soils, accounting for 0.72 to 3.94% of the total sequence reads across all gardens, sampling years, and seasons (Table 5). In a culture-based study, Berg et al. [85] found that about 3.3% of the soil bacteria isolated had antagonistic activity, which is similar to the abundance level of BCA sequence reads in this study. However, it is worth noting that certain bacterial groups are often over-represented in culture-based studies [86].

The consistent identification of extremely diverse BCA candidates (with some having the potential to be highly potent) at a great abundance level across all the gardens once again highlights the importance of the microbial community as a whole in improving soil disease suppressiveness and soil health, as has been shown previously [9,12]. The traditional approach focusing on individual antagonistic microorganisms often results in inconsistent performance and/or encounters transferability issues between cropping systems [17,87]. This is, in part, because of the fact that these microorganisms are challenged by competition from resident microbes already in soils [12]. To circumvent this limitation, a new approach, referred to as synthetic microbial communities (SynComs), constructs consortia of microorganisms to increase their adaptability and overall community stability [88,89]. It is not known at this point (1) whether the identified BCA candidates are all antagonistic against the blight pathogen, *Cps*; (2) whether and how they may work together as a consortium along with other microbes to act directly against *Cps*, and/or enhance boxwood immunity; and (3) whether these BCA candidates could be used in SynComs and work in boxwood gardens as well as other soil systems. Further investigations into these questions are warranted to harness the power of the boxwood microbiome for better plant health in production and in the landscape.

Several gardening practices may have led to the similar soil bacterial compositions across all five states/gardens. First, all gardens sampled in this study had well-established English boxwood (*Buxus sempervirens* ‘Suffruticosa’), thus similar bacterial community composition in their root zone was expected as plant–microbial relationships are often plant species-specific [90,91,92]. Second, as an iconic landscape plant, there is a set of standard cultural practices for boxwood [93]. These include the following: (1) soils must be well drained; (2) soil pH is maintained between 6.5 and 7.0; (3) soil amendment with organic matter needs to be less than 20%; and (4) the recommended fertilizer is a 12–5–9 formulation to supply N–P–K [93]. These standard practices followed by home gardeners may have favored particular bacterial groups in boxwood garden soils, as microbial composition is highly associated with soil properties [94,95]. Boxwood is also considered a low maintenance woody shrub plant once it is established [96]. This relatively low maintenance may have further contributed to the stability of the bacterial composition in these gardens and, consequently, to the improved soil suppressiveness, in contrast to intensively managed agricultural soils [97,98,99,100]. Yet, two questions of practical importance remain: (1) how applicable are the results of the present study to other private gardens; and (2) whether these results also are applicable to public gardens, which likely utilize rather different cultural practices than private gardens. The answers to these questions could have a profound impact on future plant health management in private or public gardens and landscapes.

The overall soil bacterial diversity varied across the five states and seasons, while *Cps* consistently became undetectable in those gardens [33], suggesting that bacterial diversity may not be as important as composition in soil suppressiveness. Similar observations were reported previously by Peralta et al. [24], and supported by a meta-analysis of 25 independent studies where bacterial diversity did not differ between the disease suppressive and conducive soils [8]. In this study, the observed variations in bacterial community diversity among the selected gardens and between the two seasons could be due in part to environmental factors, as shown in a previous study [101]. Soil properties, such as soil pH, moisture, and soil type, impact the physiochemistry of microorganisms colonizing the soil [102,103,104]. Other environmental factors such as precipitation [105] and temperature [106] could also have contributed to the variations seen in the soil bacterial community diversity from different locations and seasons. There is a need to elucidate the relationship between the soil microbial community and environmental factors in these gardens to materialize the findings of this study for improved boxwood health and growth.

## 5. Conclusions

By surveying soil from selected boxwood gardens, this study uncovered a variety of bacterial groups and species known to have broad biological activities ranging from nitrogen fixing to plant immunity enhancement, as well as antagonistic activity against diverse plant pathogens. These included two dominant bacterial orders—*Rhizobiales* and *Burkholderiales*—and an abundant population of 66 biological control agent candidate species. These discoveries help us to better understand the decline of the boxwood blight fungal pathogen in these garden soils, and more broadly, the low maintenance nature of boxwood as a long-lived landscape plant. This study is the first step towards harnessing the power of a garden soil microbiome or a consortium of microorganisms for improved health and growth of ornamental plants. Specifically, this study provides important leads for selecting desirable bacterial taxa that may be employed to enhance boxwood health by direct application or by improving the microbiome through strategic soil amendments or other cultural practices.

## Figures and Tables

**Figure 1 microorganisms-10-01514-f001:**
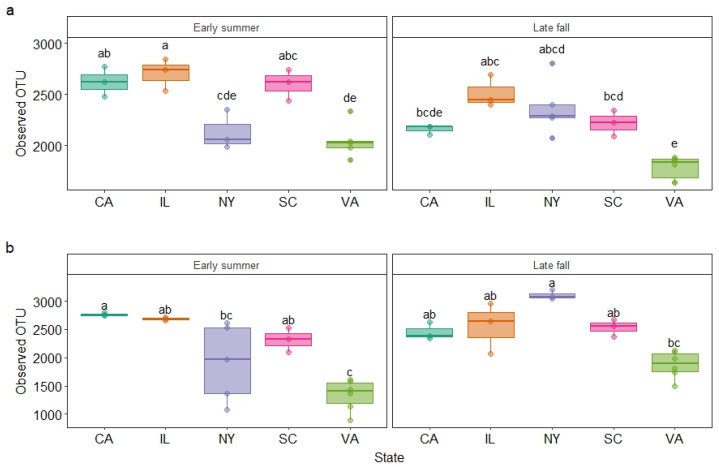
Observed OTUs of the soil samples collected from California (CA), Illinois (IL), New York (NY), South Carolina (SC), and Virginia (VA) in early summer and late summer of 2017 (**a**) and 2018 (**b**). Boxes topped by different letters within each season/year differed according to Tukey HSD post hoc test at *p* = 0.05.

**Figure 2 microorganisms-10-01514-f002:**
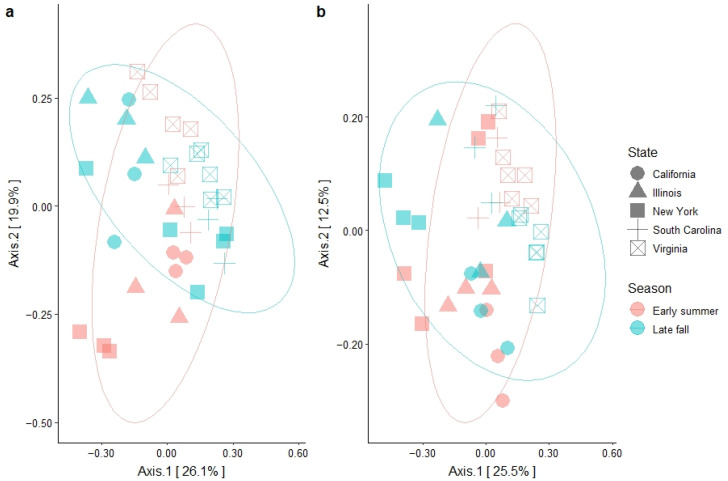
PCoA plots showing bacterial community beta diversity as a function of state and season in the 2017 (**a**) and 2018 soil samples (**b**). Sampling seasons are color coded, while the five states are indicated by different shapes. Ellipse shows the t-distribution of the season samples.

**Figure 3 microorganisms-10-01514-f003:**
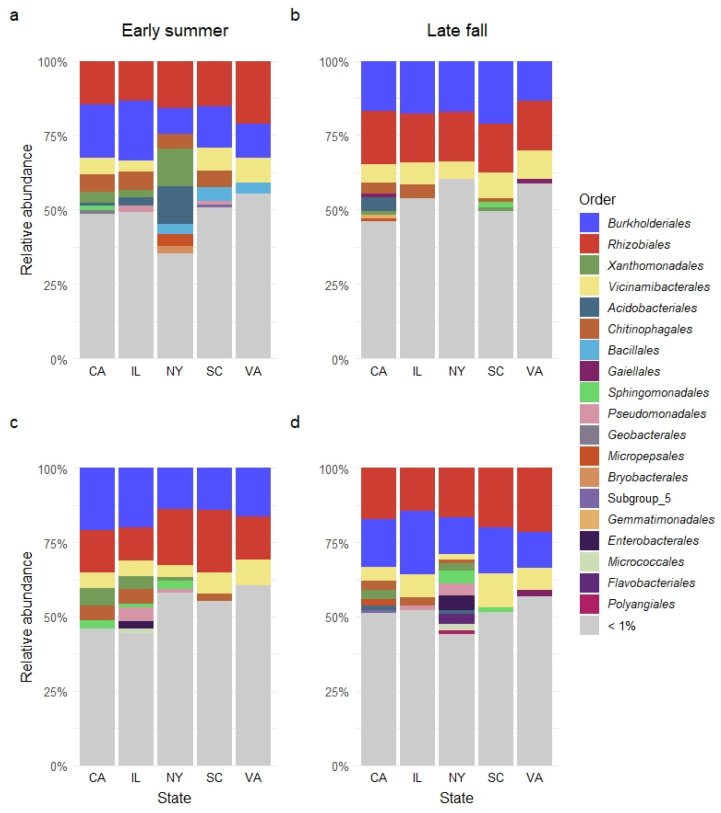
Relative abundance of soil bacterial orders identified from five gardens/states: California (CA), Illinois (IL), New York (NY), South Carolina (SC), and Virginia (VA) in early summer (**a**,**c**) and late fall (**b**,**d**) of 2017 (**a**,**b**) and 2018 (**c**,**d**). Bacterial orders with relative abundance less than 1% were grouped together and indicated by gray color.

**Figure 4 microorganisms-10-01514-f004:**
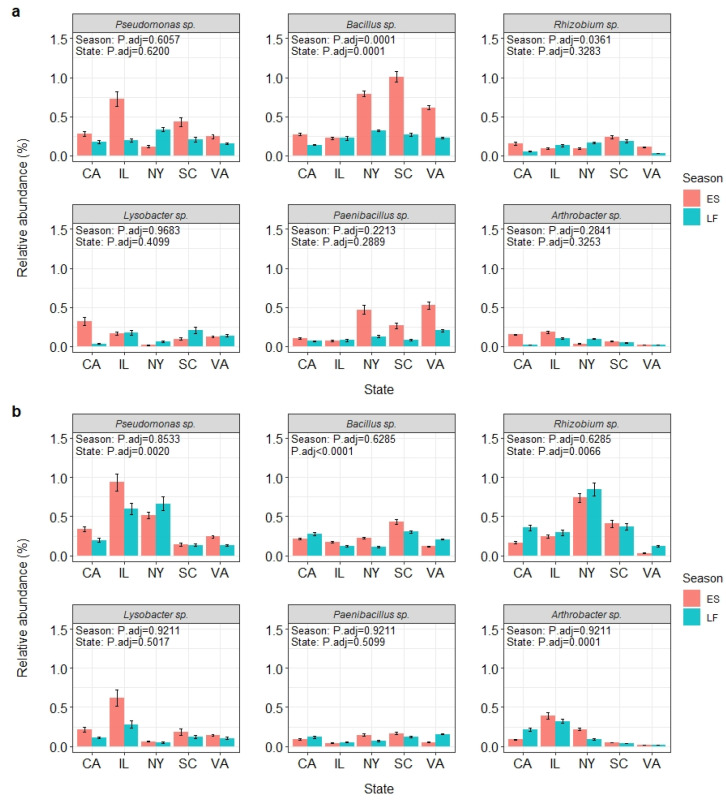
Kruskal–Wallis test on the relative abundance of the six most abundant bacterial BCA candidate species found from the California (CA), Illinois (IL), New York (NY), South Carolina (SC), and Virginia (VA) samples collected in early summer (ES) and late fall (LF) of 2017 (**a**) and 2018 (**b**).

**Table 1 microorganisms-10-01514-t001:** Summary of quality bacterial sequence reads and sample coverage by year and season.

Year	Season	Sequence Reads	Sample Coverage
2017	Early summer	1,788,962	0.991
	Late Fall	2,127,058	0.991
2018	Early summer	1,871,473	0.995
	Late Fall	1,732,272	0.990
∑		7,519,765	Avg = 0.992

**Table 2 microorganisms-10-01514-t002:** Analysis of variance on two alpha diversity indices among five states and between two seasons and their interactions by year.

Year	Variable	Observed OTU Richness		Shannon Index	
		F	*p*-Value	F	*p*-Value
2017	State	20.60	<0.0001	27.72	<0.0001
	Season	12.97	0.0013	5.45	0.0273
	Season × State	4.53	0.0063	8.56	0.0001
2018	State	15.06	<0.0001	31.35	<0.0001
	Season	11.72	0.0019	0.54	0.4704
	Season × State	5.42	0.0023	4.23	0.0084

Significant *p*-values are shaded in gray.

**Table 3 microorganisms-10-01514-t003:** PERMANOVA and dispersion analyses of bacterial beta diversity in soil samples collected from the five states during early summer and late fall of each year.

Year	Variable	PERMANOVA ^†^		Dispersion ^‡^	
		R^2^	*p*-Value	F	*p*-Value
2017	State	0.33	<0.0001	5.07	0.0028
	Season	0.06	0.0006	0.41	0.5268
	State × Season	0.22	<0.0001	- ^§^	-
2018	State	0.38	<0.0001	4.89	0.0033
	Season	0.05	0.0026	0.45	0.5063
	State × Season	0.13	0.0004	-	-

^†^: Permutational multivariate analysis of variance at 10,000 permutations; ^‡^: analysis of multivariate homogeneity of group variances; ^§^: not tested; significant *p*-values are shaded in gray.

**Table 4 microorganisms-10-01514-t004:** Kruskal–Wallis test on the relative abundance of the ten most abundant bacterial orders among the five states/gardens by season and year and between two seasons by state and year.

Order	Relative Abundance (%)	Season				State			
		2017		2018		2017		2018	
		χ^2^	*p*-Adj	χ^2^	*p*-Adj	χ^2^	*p*-Adj	χ^2^	*p*-Adj
*Rhizobiales*	16.8	1.64	0.5751	3.84	0.7027	9.59	0.0741	15.63	0.0117
*Burkholderiales*	16.0	0.02	0.9482	0.31	0.9372	21.70	0.0020	22.51	0.0022
*Vicinamibacterales*	6.9	1.34	0.6096	0.003	0.9893	15.52	0.0103	19.41	0.0041
*Chitinophagales*	3.7	5.80	0.2555	2.78	0.7787	18.55	0.0043	20.48	0.0032
*Xanthomonadales*	2.8	2.81	0.4531	1.17	0.8771	18.23	0.0048	26.25	0.0000
*Gaiellales*	1.9	0.06	0.9288	9.24	0.2309	21.51	0.0021	9.56	0.0812
*Sphingomonadales*	1.8	0.68	0.6814	1.30	0.8643	25.69	0.0011	24.69	0.0015
*Bacillales*	1.8	10.44	0.0993	0.09	0.9836	1.65	0.8147	14.05	0.0184
*Acidobacteriales*	1.7	3.92	0.3257	0.77	0.9372	13.58	0.0197	13.40	0.0224
*Gemmatimonadales*	1.6	0.004	0.9715	0.01	0.9893	14.99	0.0124	15.26	0.0131

Significant *p*-values are shaded in gray.

**Table 5 microorganisms-10-01514-t005:** Total abundance of 66 bacterial species with potential biological control activities against plant pathogens by year, season, and garden/states.

Year	Season	State	Total BCA Sequence Reads	Total Sequence Reads	BCA Abundance (%)
2017	Early summer	California	6893	409,612	1.68
		Illinois	8048	396,132	2.03
		New York	5529	315,309	1.75
		South Carolina	8077	315,116	2.56
		Virginia	7514	352,793	2.13
	Late fall	California	1266	174,912	0.72
		Illinois	1503	136,271	1.10
		New York	8937	668,827	1.34
		South Carolina	5590	467,264	1.20
		Virginia	7254	679,784	1.07
2018	Early Summer	California	7313	429,281	1.70
		Illinois	13,630	346,078	3.94
		New York	9419	386,764	2.44
		South Carolina	4166	249,726	1.67
		Virginia	4330	459,624	0.94
	Late fall	California	6020	349,274	1.72
		Illinois	5644	251,778	2.24
		New York	4282	143,548	2.98
		South Carolina	2933	215,495	1.36
		Virginia	8226	772,177	1.07

## Data Availability

The data presented in this study are available at the NCBI SRA archive (BioProject: PRJNA835795); The R codes used in this study are available at https://github.com/lixiaopi1985/garden-soil-microbiome.git (accessed on 9 May 2022).

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
