# Peer review of "Characterization of the Soil Bacterial Community from Selected Boxwood Gardens across the United States"

_microorganisms, 2022, doi:10.3390/microorganisms10081514_

Round 1

Reviewer 1 Report

The manuscript examined the bacterial communities in five boxwood garden soils demonstrated to be suppressive to the box wood blight pathogen Calonectria pseudonaviculata (Cps). Overall, the approach of the study is good and could be useful in the public domain, but the manuscript needs considerable revision to reach the public domain. Authors are suggested to address following comments in order to make the manuscript suitable for publication.

Abstract should be rewritten by detailing the aim and concept of the study. The abstract should state briefly the purpose of the study, the principal results and major conclusions.

*Provide significant words which are more relevant to the work in logical sequence as ‘keywords’.

* Introduction is very general and need to be elaborative to explore the actual philosophy to design the experiment. The introduction is insufficient to provide the state of the art in the topic. The originality and novelty of the paper need to be further clarified. What progress against the most recent state-of-the-art similar studies was made in this study?

*The introduction of the paper must be extended and reformulated in order to provide a more comprehensive approach.

* Improve the introduction part by adding more supporting data from the literature

* Given the details of DNA extraction from the sample, exactly how much sample was taken for DNA extraction? Was it dried sample or fresh sample?

* It would be necessary to develop more bioinformatic/statistical analyses in the present study. Moreover, did the authors upload the sequences into a datable? Please give the accession number in the manuscript.

* In the materials and methods section; provide the sampling coordinates.

*The manuscript does not provide interesting and technically sound discussion; it would be better to use more recent references in discussion.

*Under section, discussion, it is recommended to discuss and explain what should be the appropriate policies based on the findings of this study. Also, the results should be further elaborated to show how they could be used for real applications. 

*Authors are suggested to draw major inferences/primary conclusions first quoting the data/results obtained followed by the secondary conclusions/ recommendations reached through the critical analysis/ investigation of the study. Based on the outcome of the study, the author(s) may recommend the extension of the present study as the future scope of research.

Reviewer 2 Report

Xiaoping Li et al's manuscript "The Soil Bacterial Community in Five Boxwood Gardens across 2 the United States" explored the resistant microbiomes of five garden soils. The paper is well written. However, the experiment design and the data analysis could be improved. I think the research is worth to be published once revised. I have detailed comments in the attachment.  

Round 2

Reviewer 1 Report

The authors presented a satisfactory revision to previous comments. However, there are further queries that need to be addressed before publication

*Results and discussion: Provide a better explanation for your data. Avoid only comparing your results to previous studies. Interpret and discuss the meaning of your results more deeply.  Discussions need to be supported by the latest references and need to be explained in depth. The authors should highlight the reason of their result findings in the light of available literature.

* Conclusion: Conclusions is not just about summarizing the key results of the study, it should highlight the insights and the applicability of your findings/results for further work.  Please enrich your conclusion.
